# Effect of experimental hookworm infection on insulin resistance in people at risk of type 2 diabetes

Doris R. Pierce[1], Malcolm McDonald[1], Lea Merone[2], Luke Becker [1],
Fintan Thompson[1,3], Chris Lewis[2], Rachael Y. M. Ryan[1], Sze Fui Hii[4],
Patsy A. Zendejas-Heredia [4], Rebecca J. Traub[4], Matthew A. Field [1,5,6],
Tony Rahman [7], John Croese[1], Alex Loukas[1], Robyn McDermott[1,3] &
Paul R. Giacomin [1] ✉

The reduced prevalence of insulin resistance and type 2 diabetes in countries with endemic parasitic worm infections suggests a protective role for worms against metabolic disorders, however clinical evidence has been non-existent. This 2-year randomised, double-blinded clinical trial in Australia of hookworm infection in 40 male and female adults at risk of type 2 diabetes assessed the safety and potential metabolic benefits of treatment with either 20 ($n = 14$) or 40 ($n = 13$) *Necator americanus* larvae (L3) or Placebo ($n = 13$) (Registration ACTRN12617000818336). Primary outcome was safety defined by adverse events and completion rate. Homoeostatic model assessment of insulin resistance, fasting blood glucose and body mass were key secondary outcomes. Adverse events were more frequent in hookworm-treated participants, where 44% experienced expected gastrointestinal symptoms, but completion rates were comparable to Placebo. Fasting glucose and insulin resistance were lowered in both hookworm-treated groups at 1 year, and body mass was reduced after L3-20 treatment at 2 years. This study suggests hookworm infection is safe in people at risk of type 2 diabetes and associated with improved insulin resistance, warranting further exploration of the benefits of hookworms on metabolic health.

Metabolic syndrome (MetS) represents a collection of related metabolic conditions, including dysregulated lipid homoeostasis, hypertension, insulin resistance, and abdominal obesity, that increase the likelihood of developing chronic diseases such as type 2 diabetes mellitus (T2D) and cardiovascular disease[1]. T2D is a complex illness associated with hyperglycaemia and insulin resistance, where long-term and costly medical care is usually required to prevent complications. The prevalence of T2D is increasing globally, with more than half a billion adults already diagnosed[2]; hence there is a pressing need to develop new preventative approaches to limit the burden of T2D on individuals and health systems.

The soaring global prevalence of T2D is undoubtedly related to lifestyle factors such as an energy-dense diet and a sedentary lifestyle. In this context, the detrimental role of systemic, low-grade

[1]Centre for Molecular Therapeutics, Australian Institute of Tropical Health and Medicine, James Cook University, Cairns, QLD, Australia. [2]College of Health Sciences, James Cook University, Cairns, QLD, Australia. [3]University of South Australia, Adelaide, SA, Australia. [4]Melbourne Veterinary School, Faculty of Science, University of Melbourne, Parkville, VIC, Australia. [5]College of Public Health, Medical & Vet Sciences, James Cook University, Cairns, QLD, Australia. [6]Immunogenomics Laboratory, Garvan Institute of Medical Research, Darlinghurst, NSW, Australia. [7]The Department of Gastroenterology and Hepatology, The Prince Charles Hospital, Brisbane, QLD, Australia. ✉e-mail: paul.giacomin@jcu.edu.au

inflammation in promoting insulin resistance is well-established[3]. In lean adipose tissue, there exists an anti-inflammatory environment, including regulatory T cells, eosinophils, and alternatively activated macrophages that are stimulated by type 2 cytokines such as interleukin (IL)−4, IL-5, and IL-13. Conversely, in obesity there is a pro-inflammatory milieu involving type 1 cytokines such as IFNγ and TNF, classically-activated macrophages[4], and a dysregulated gut microbiome[5]. This biased Type-1 immune response aggravates insulin resistance through stress-activated kinases that target intermediates in the insulin-signalling pathway[6]. It follows that maintaining the systemic and adipose-specific inflammatory balance in favour of type 2 or regulatory immune response is a rational approach to limiting the inflammatory cascade and preventing the ensuing insulin resistance seen in metabolic disease.

Parasitic helminths (worms) co-evolved with humans and remain endemic in regions of the world with less metabolic and inflammatory diseases[7–10]. Helminth infections are associated with type 2 immune responses such as eosinophilia, elevated IL-4, −5, and −13, Type-2 innate lymphoid cell responses, and modulation of the gut microbiome[11]. Emerging evidence supports the hypothesis that eradicating worms from industrialised regions may have partly contributed to the increased prevalence of immune-mediated diseases such as T2D[8,12]. For example, human epidemiological studies suggest that the presence of helminth infections is associated with lessened inflammatory responses and improved glucose homoeostasis[10,12,13], and two systematic reviews and meta-analyses concluded that adults with previous or current worm infection typically enjoy improved metabolic function[14,15]. While cross-sectional studies support the idea of a protective role for worms in metabolic disease, they did not establish causality.

Recent deworming studies provided some causative evidence. The removal of worms from previously-infected individuals was associated with increased insulin resistance (elevated homoeostatic model assessment for insulin resistance, HOMA-IR)[16], elevated serum insulin and glucagon[17], enhanced hyperglycaemia and risk of metabolic disease[18], and worsened blood lipid profiles[19]. Experimental mouse studies corroborate findings from human studies, where infection with various worm species caused improvements in metabolic indices, type 2 immune responses, and alterations to the gut microbiome[20–24]. However, to date, there have been no studies in humans testing whether a controlled helminth infection is safe, can improve metabolic health, and potentially prevent progression to T2D.

Previously, we and others have established the safety of infecting human volunteers with low doses of hookworms in various inflammatory diseases[25–27]. While none of these studies investigated metabolic health outcomes, it is well reported that hookworms elicit type 2 and regulatory immune responses[26] and changes in the microbiome[28] that may be beneficial in the context of T2D. The primary aim of this randomised, double-blinded, placebo-controlled Phase 1b clinical trial was to establish the safety, tolerability, and potential metabolic benefits of experimental infection with infective third-stage larvae (L3) of the human hookworm *Necator americanus* in otherwise healthy adults with metabolic syndrome and at risk of developing T2D. Based on previous animal and deworming studies, we hypothesised that hookworm infection would be safe and improve the key metabolic outcome for the study, HOMA-IR, which assesses longitudinal changes in insulin sensitivity that correlate well with estimates derived via the euglycaemic clamp[29]. In this study we demonstrate that hookworm infection is safe and well-tolerated in people at risk of type 2 diabetes, and provide proof of principle that hookworm treatment may stabilise or improve key determinants of metabolic health such as insulin resistance.

## Results
### Study population
We conducted a 2-year Phase Ib randomised, double-blinded, three-arm placebo-controlled trial of experimental hookworm infection (either 20 or 40 infective third-stage larvae, L3) in otherwise healthy people at risk of T2D (Fig. 1A). Recruiting progressed between January 2018 and June 2020, with 85 potential participants meeting screening criteria, of which 41 met the inclusion/exclusion criteria to proceed to randomisation (4 short of the recruitment target). Forty-four participants were excluded due to not having elevated baseline HOMA-IR ($n = 27$), not responding to communication ($n = 11$) or excluded medications or medical conditions ($n = 6$) (Fig. 1B). One participant who qualified for the study dropped out prior to their baseline visit. Forty participants were randomised and constituted the intention-to-treat (ITT) population (Table 1). Thirteen participants were allocated to receive Placebo treatment, 14 to receive a dose of 20 hookworm larvae (L3-20) and 13 to receive a dose of 40 hookworm larvae (L3-40). (Fig. 1B). The majority of participants (85%) were of Caucasian descent. Baseline age and HOMA-IR were higher in the L3-20 group. One participant who was randomised into the Placebo group unintentionally received an L3-20 treatment due to a labelling error. This participant was included in the Placebo group for the ITT analysis but in the L3-20 group for per-protocol analyses.

### Adverse events (AEs)
The study's primary outcome was the safety and tolerability of hookworm treatment, assessed by an intention to treat analysis of the frequency and severity of AEs. A total of 20 AEs and no serious AEs were reported (Table 2). As expected, gastrointestinal (GI) adverse events occurred in the L3-20 and L3-40 treatment groups (Table 2), consistent with the worms arriving in the gut and attaching to the intestinal wall. These AEs were experienced by 44% of hookworm-treated participants and included bloating, nausea, vomiting, constipation, diarrhoea, epigastric upset, hungry feeling, stomach cramps and abdominal pain, which were typically *mild-moderate* and resolved without medical intervention. One GI-related symptom (5%) was recorded as *severe*, and three warranted early removal of the participants from the study (1 in the L3-20 group, 2 in the L3-40 group) and provision of deworming medication. Two of these individuals recovered promptly after deworming, while symptoms in one participant persisted, indicating their symptoms may have been unrelated to hookworm. No GI-related AEs occurred in the Placebo group. No participant developed anaemia (Supplementary Fig. 1), the primary clinical effect of moderate to heavy hookworm infection. AEs deemed unrelated to worm infection were reported across all groups and included one case each of concussion, laryngitis, appendicectomy, gynaecological procedure, salmonella infection acquired overseas, breast cancer diagnosis, erythema, and dermatitis. The chi-square test for trend revealed a significant linear trend for AEs, $\chi^2 (1, N = 40) = 3.846, p = 0.049$, with more AEs occurring in the hookworm treatment groups compared to the Placebo group, as anticipated based on studies in other patient populations[30].

### Study Progression and well-being
Rates of trial completion and participant mood and well-being were safety outcomes, which were all conducted using per protocol analyses. The CONSORT Flow Diagram (Fig. 1B) and Kaplan−Meier curve (Fig. 1C) detail study progression. Early dropouts (0-6 months) included the 3 participants with GI symptoms and one participant in the L3-20 group that was removed from the study at 3 months due to an unrelated cancer diagnosis (Fig. 1B). Reasons for early terminations later in the study included 4 participants moving away (Placebo $n = 2$, L3-20 $n = 1$, L3-40 $n = 1$), 1 for undisclosed personal reasons (L3-20), 3 for failure to respond to communication or testing exhaustion (Placebo $n = 2$, L3-40 $n = 1$), and 4 for starting medication (L3-20, $n = 1$) or undergoing gastric sleeve surgery (Placebo $n = 1$, L3-20 $n = 2$) that could interfere with the trial outcomes. Four participants (Placebo $n = 1$, L3-20 $n = 2$, L3-40 $n = 1$) missed one evaluation visit each but completed the remaining evaluation visits. Twenty-four of the 40 randomised participants completed the trial, including 8 (62%) in the

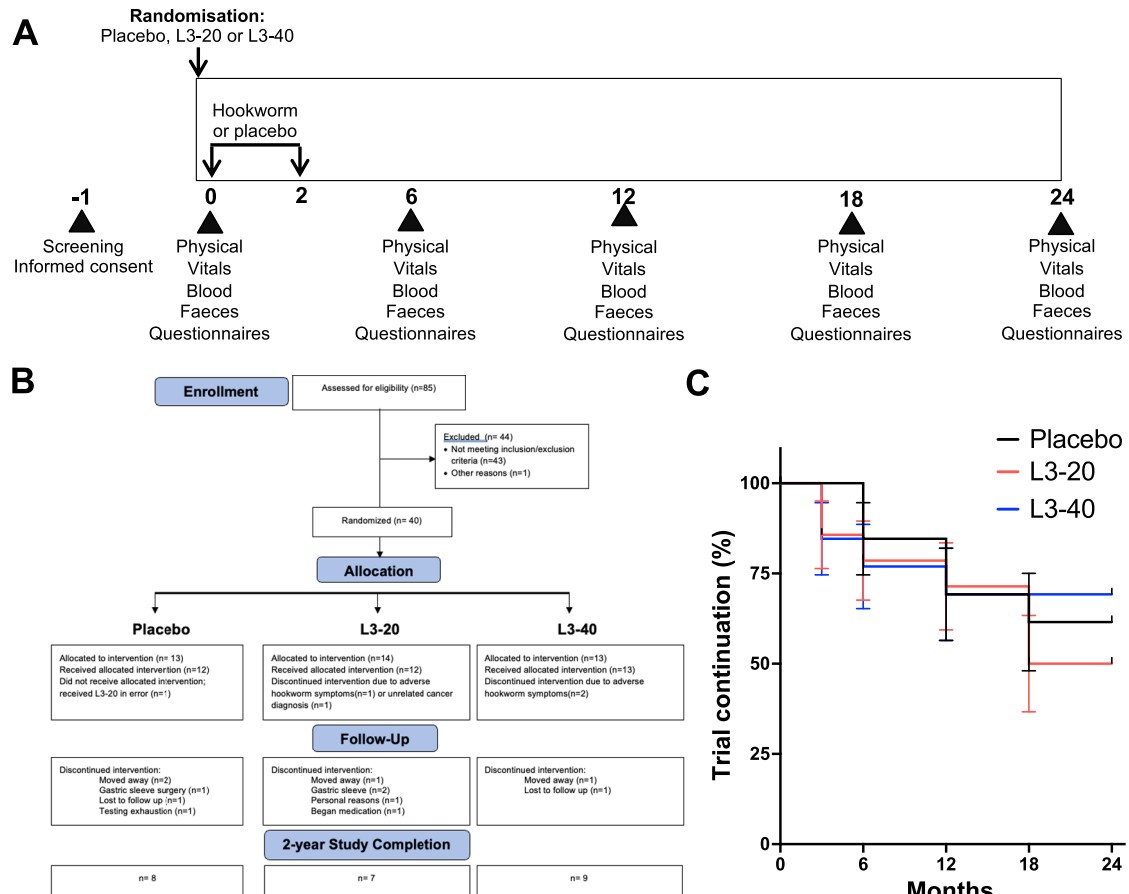

**Fig. 1 | Study time line, CONSORT flow diagram and study progression. A** After screening and informed consent, participants were allocated to receive either placebo (chilli pepper solution) or a total of 20 (L3-20) or 40 (L3-40) *N. americanus* hookworm larvae delivered to the skin over 2 occasions, at month 0 and month 2. Participants underwent evaluation visits every 6 months, where physical and vitals exams were undertaken, adverse events were reviewed, and biological samples and questionnaires were collected for analysis of safety, pathology, well-being, diet and exercise habits. **B** CONSORT chart showing flow of patients through the clinical trial, and reasons for early termination. **C** Kaplan–Meier analysis of rates of trial continuation (percentage survival and standard error of the mean) in each group during the 2-year study (Placebo *n* = 13, L3-20 *n* = 14, L3-40 *n* = 13; Log-rank test for trend, *p* = 0.832). Source data are provided as a Source Data File.

Placebo, 7 in the L3-20 (50%) and 9 in the L3-40 group (69%). The difference in study completion between groups was not significant (Fig. 1C, Log-rank test for trend, *p* = 0.832). Monitoring of participant mood and depressive state via the Patient Health Questionnaire-9 (PHQ-9) revealed that median scores in the Placebo group remained relatively stable throughout (Table 3). There was a trend for reduced median PHQ-9 scores (suggestive of improved mood) compared to baseline in both hookworm treatment groups at all evaluation visits; however, longitudinal and inter-cohort analyses did not detect any significant differences over time or between groups, even when the active hookworm treatment groups were combined (Mann–Whitney test) (Table 3).

### Analysis of potential confounding variables: diet and exercise habits

Participants completed regular questionnaires to record any changes in diet and exercise habits that could impact metabolic health results whilst undergoing treatment during the trial[31]. Analysis of the diet

### Table 1 | Intention to treat baseline characteristics for all randomised participants (*n* = 40)

|  | Allocation | | |
|---|---|---|---|
|  | Placebo (*n* = 13) | L3-20 (*n* = 14) | L3-40 (*n* = 13) |
| Median age (range), years | 35 (27–50) | 43 (30–45) | 36 (27–49) |
| Female, *n* (%) | 10 (75%) | 10 (73%) | 10 (77%) |
| Body mass, median (IQR), kg | 106 (90–121) | 104 (89–126) | 102 (90–130) |
| Height, median (IQR), cm | 168 (162–176) | 164 (158–174) | 168 (165–178) |
| BMI, median (IQR), kg/m² | 37 (33–41) | 38 (33–46) | 36 (32–41) |
| Waist Circumference, median (IQR), cm | 111 (102–122) | 111 (103–116) | 111 (105–123) |
| HOMA-IR, median (IQR), units | 2.2 (2.0–2.8) | 3.0 (2.3–3.5) | 2.4 (2.0–3.2) |

### Table 2 | Summary of the number of adverse events (AE) and proportions of participants experiencing at least one AE, classified by AE type and symptom severity

|  | Total number AEs (Proportion with > 1 AE) | | | |
|---|---|---|---|---|
|  | Placebo (*n* = 13) | L3-20 (*n* = 14) | L3-40 (*n* = 13) | Combined hookworm (*n* = 27) |
| Any AE | 3 (15%) | 9 (50%) | 8 (46%) | 17 (48%) |
| Unrelated | 3 (15%) | 4 (21%) | 1 (8%) | 5 (19%) |
| Hookworm-related (GI), and severity | 0 | 5 (36%) | 7 (54%) | 12 (44%) |
|  |  | Mild: 1 | Mild: 2 | Mild: 3 |
|  |  | Moderate: 4 | Moderate: 4 | Moderate: 8 |
|  |  | Severe: 0 | Severe: 1 | Severe: 1 |

questionnaire (PREDIMED) revealed no significant inter-cohort differences or longitudinal changes in PREDIMED score in any treatment group (Supplementary Fig. 2A). Similarly, analysis of the exercise habits of participants by quantifying their median numbers of metabolic equivalent of tasks (METs) per week revealed no significant changes within groups from baseline and no significant differences in median absolute METs/week or changes in METs/week between groups at any time point (Supplementary Fig. 2B).

**Evaluation of hookworm establishment and immune responses**
Hookworm infection is associated with the presence of hookworm eggs in stool, peripheral blood eosinophilia and a type 2 cytokine response. As expected, eosinophil counts remained stable in the Placebo cohort (Fig. 2A), and faecal samples were negative for hookworm ova (Fig. 2C). In both the L3-20 and L3-40 groups, eosinophil counts peaked and were significantly elevated from baseline at 6 months (L3-

20 $p = 0.047$, L3-40 $p = 0.002$, Tukey's posthoc test) but began decreasing back to normal values after 12 months (Fig. 2A). Of the 28 hookworm-treated participants, five faecal samples were unavailable for qPCR testing due to early dropout or inability to deliver a faecal sample. Twenty-one of the 23 participants (91%) tested positive for *N. americanus* ova at 6 months (Fig. 2C), and the two participants that tested negative still displayed robust increases in eosinophil counts (Fig. 2B). Quantification of hookworm infection intensity (hookworm eggs per gram, EPG) revealed that there was substantial variability in hookworm EPGs between hookworm-treated participants, with no clear differences in infection intensity between groups that received either 20 or 40 hookworm L3 (Fig. 2D). Analysis of serum cytokine responses in the combined hookworm cohorts revealed significantly increased levels of IL-5 at both 6 and 12 months ($p = 0.0001$ and $p = 0.02$, respectively); however levels of other type 2 or regulatory cytokines such as IL-4, IL-13 and IL-10 were not significantly elevated

**Table 3 | Patient Health Questionnaire scores in each treatment group**

| Patient Health Questionnaire (PHQ)-9 scores, Median and IQR (n) | | | | | |
|---|---|---|---|---|---|
| | Baseline | 6 months | 12 months | 18 months | 24 months |
| Placebo | 7 | 5 | 6 | 6 | 9 |
| | 4–14 (n = 12) | 3–8 (n = 12) | 1–13 (n = 9) | 3–9 (n = 8) | 2–16 (n = 7) |
| L3-20 | 8 | 4 | 4 | 3 | 3 |
| | 2–11 (n = 13) | 2–7 (n = 12) | 2–7 (n = 11) | 1–9 (n = 11) | 1–6 (n = 8) |
| L3-40 | 10 | 9 | 5 | 7 | 4 |
| | 5–12 (n = 11) | 3–12 (n = 11) | 3–8 (n = 10) | 2–10 (n = 8) | 2–9 (n = 9) |
| Combined Hookworm | 8 | 5 | 4 | 6 | 4 |
| | 3–12 (n = 24) | 2–11 (n = 23) | 2–7 (n = 21) | 1–9 (n = 19) | 2–8 (n = 17) |

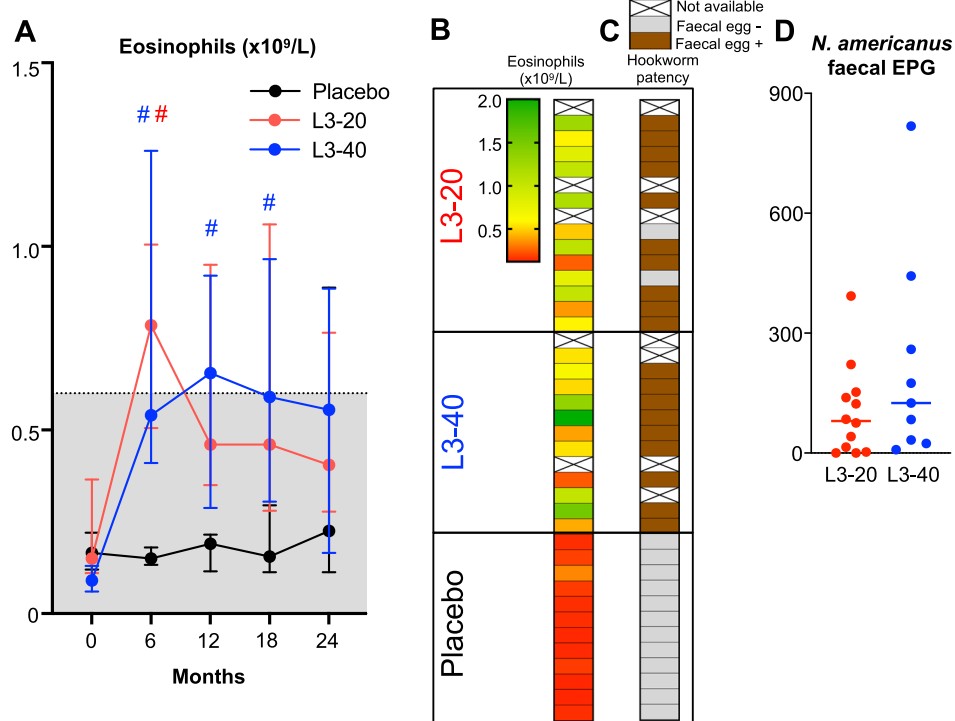

**Fig. 2 | Evaluation of establishment of hookworm infections. A** Peripheral blood eosinophil counts (median ± IQR) at each evaluation visit, shaded area indicates normal range (Sample size for month 0, 6, 12, 18 and 24: Placebo n = 12, 12, 9, 8 and 6; L3-20 n = 13, 12, 11, 11 and 8; L3-40 n = 11, 11, 10, 8 and 8, respectively). # Indicates significantly different to 0 months; Tukey's posthoc test: (**B**) Heat map displaying eosinophil counts the 6 month visit (peak eosinophilia) in each participant and correlation with the (**C**) detection of hookworm eggs in faeces by qPCR (Hookworm patency). Some samples were unable for analysis due to dropping out prior to the 6-month visit. **D** Quantification of hookworm eggs per gram (EPG) in faecal samples at 6 months (median, and individual data points, L3-20 n = 12, L3-40 n = 9). Source data are provided as a Source Data File.

above baseline (Supplementary Fig. 3). Similarly, there were no significant alterations in the levels of pro-inflammatory cytokines such as IFNγ, TNF, or IL-2; however IL-17A levels were significantly reduced at 6 months after treatment ($p = 0.03$, Supplementary Fig. 3).

## Metabolic outcomes: Insulin resistance (HOMA-IR)

We undertook a per-protocol analysis of secondary metabolic outcome measures to account for pre-established protocol deviations, including the Placebo participant who received hookworm treatment in error. Data collected from the 4 participants described in the Study Progression section who, at 18 or 24 months, were retrospectively discovered to have undertaken gastric sleeve surgery or began taking metformin were also excluded from the analysis. Data from one L3-20 participant whose insulin (47 mU/L), glucose (12.4 mmol/L) and HOMA-IR (26 units) values at 6 months were well beyond normal ranges and inconsistent with fasting were also excluded.

A key secondary outcome of the trial was to determine the effect of hookworm treatment versus placebo on insulin resistance, as assessed by changes in HOMA-IR. Median HOMA-IR values in the Placebo group fluctuated between improved and worsened from the baseline value of 2.2 [2.0–2.6], reaching a maximum of 2.9 [1.8–4.2] at 12 months, corresponding to a median increase of 0.8 units [−0.7 to 2.3] (Table 4 and Fig. 3A). In contrast, median HOMA-IR value in the L3-20 group was significantly lowered from the baseline value of 3.0 [2.3–3.6] to 1.8 [1.7–2.5] at 12 months ($p = 0.039$) and 2.1 [1.5–2.3] at 24 months ($p = 0.047$). Median change in HOMA-IR values in the L3-20 group was significantly different from Placebo at 12 months ($p = 0.039$, Kruskal–Wallis). Similar trends toward lowered HOMA-IR values were observed in the L3-40 group at 6 months (−0.5 units [−1.2 to 0.2]) and 12 months (−0.4 units [−1.4 to 0.9]); however, these changes were not statistically different from Placebo at any evaluation visit. To investigate whether improvements in HOMA-IR were correlated with increased hookworm infection intensity, we analysed changes in

HOMA-IR at 6 months relative to hookworm eggs per gram (EPG). Simple linear regression analysis revealed no association between infection intensity and improvements in insulin resistance ($R^2 = 0.0005$, $p = 0.93$, Supplementary Fig. 4).

Since significance tests depend greatly on sample size, and numbers in all groups were low, we calculated the effect sizes of treatment versus control to measure the treatment effect independent of participant numbers. Effect sizes for HOMA-IR changes in the L3-20 versus Placebo group were large at 6 months ($d = 0.80$), 12 months ($d = 1.34$), 18 months ($d = 1.19$) and 2 years ($d = 0.90$). For the L3-40 group, effect sizes were small at 6 months ($d = 0.48$), medium at 12 months ($d = 0.73$), negligible at 18 months ($d = 0.18$), and small at 2 years ($d = −0.40$) (Supplementary Table 1).

## Metabolic outcomes: fasting blood glucose, insulin and glycated haemoglobin

Median FBG levels (mmol/L) in the Placebo group were stable throughout and did not significantly deviate from their starting values ($p = 0.605$, Durbin, Skillings–Mack test) (Table 4 and Fig. 3B). In contrast, median FBG levels were reduced from baseline levels after 6 months in both the L3-20 (baseline: 5.2 [4.7–5.6] to 6 months: 4.5 [4.0–5.1]; $p = 0.016$) and L3-40 groups (baseline: 5.3 [4.9–5.4] to 6 months: 4.3 [4.0–5.3]; $p = 0.012$). Similar significant reductions in FBG were seen in the L3-40 group at 12 months ($p = 0.012$), 18 months ($p = 0.047$) and 24 months ($p = 0.012$) compared to baseline; however, reductions in the L3-20 group were not statistically significant at any other time point. Median changes in FBG levels in the L3-20 treatment group were not significantly different from Placebo at any time point, but changes in the L3-40 group were significantly different to Placebo at 6 months (−0.7 in L3-40 compared to 0.0 in Placebo; $p = 0.027$) and 12 months (−1.0 in L3-40 compared to +0.3 in Placebo; $p = 0.024$).

Fasting insulin levels (mU/L) were stable in the Placebo group throughout the study, ranging from a baseline of 11 [9–12] to a

## Table 4 | Absolute values and changes in HOMA-IR, fasting blood glucose and insulin in each treatment group

| HOMA-IR (units), Median and IQR (n) | | | | | | | | |
|---|---|---|---|---|---|---|---|---|
| | Baseline | 6 months | Δ6 mo. | 12 months | Δ12 mo. | 18 months | Δ18 mo. | 24 months | Δ24 mo. |
| Placebo | 2.2 | 2.6 | −0.2 | 2.9 | +0.8 | 2.0 | −0.2 | 2.0 | −0.3 |
| | 2.0–2.6 (n = 12) | 2.1–3.1 (n = 12) | −0.6 to 1.0 | 1.8–4.2 (n = 9) | −0.7 to 2.3 | 1.6–3.8 (n = 8) | −0.9 to 2.3 | 1.4–2.7 (n = 6) | −0.8 to 0.6 |
| L3-20 | 3.0 | 2.3 | −0.6 | **1.8ᵃ** | **−1.1ᵇ** | 1.6 | −1.5 | **2.1ᵃ** | −0.8 |
| | 2.3–3.5 (n = 13) | 1.5–3.2 (n = 11) | 1.9 to −0.1 | 1.7–2.5 (n = 11) | −1.5 to −0.5 | 0.9–3.0 (n = 8) | −1.8 to −0.1 | 1.5–2.3 (n = 7) | −1.3 to −0.4 |
| L3-40 | 2.4 | 2.1 | −0.5 | 2.4 | −0.4 | 2.5 | −0.3 | 1.5 | −0.5 |
| | 2.1–3.0 (n = 11) | 1.8–2.6 (n = 11) | −1.2 to 0.2 | 1.7–3.2 (n = 10) | −1.4 to 0.9 | 1.9–3.1 (n = 8) | −0.4 to 0.4 | 1.0–4.8 (n = 9) | −1.2 to 1.1 |
| Fasting blood glucose (mmol/L), Median and IQR (n) | | | | | | | | |
| Placebo | 4.8 | 5.0 | 0.0 | 4.8 | +0.3 | 4.6 | −0.1 | 4.0 | −0.8 |
| | 4.5–5.1 (n = 12) | 4.0–5.3 (n = 12) | −0.5 to 0.5 | 4.4–5.4 (n = 9) | −0.4 to 0.8 | 4.2–5.3 (n = 8) | −0.7 to 0.6 | 3.1–5.1 (n = 6) | −1.9 to 0.6 |
| L3-20 | 5.2 | **4.5ᵃ** | −0.7 | 4.2 | −0.9 | 4.5 | −0.5 | 4.7 | −0.6 |
| | 4.7–5.6 (n = 13) | 4.0–5.1 (n = 11) | −1.0 to −0.3 | 3.6–5.6 (n = 11) | −1.1 to 0.6 | 4.2–4.7 (n = 8) | −1.0 to 0.1 | 3.9–4.8 (n = 7) | −1.2 to −0.4 |
| L3-40 | 5.3 | **4.3ᵃ** | **−0.7ᵇ** | **4.1ᵃ** | **−1.0ᵇ** | **4.5ᵃ** | −0.4 | **4.3ᵃ** | −0.6 |
| | 4.9–5.4 (n = 11) | 4.0–5.3 (n = 11) | −1.0 to −0.6 | 3.7–4.9 (n = 10) | −1.4 to −0.7 | 4.1–5.1 (n = 8) | −1.2 to 0.0 | 4.0–4.9 (n = 9) | −1.1 to -0.4 |
| Fasting blood insulin (mmol/L), Median and IQR (n) | | | | | | | | |
| Placebo | 11 | 12 | +1 | 12 | +2 | 11 | 0 | 11 | +1 |
| | 9–12 (n = 12) | 10–15 (n = 12) | −2 to 4 | 8–19 (n = 9) | −3 to 10 | 8–6 (n = 8) | −3 to 8 | 10–14 (n = 6) | −1 to 2 |
| L3-20 | 13 | 10 | −2 | **10ᵃ** | **−4ᵇ** | 9 | −5 | 10 | −3 |
| | 12–16 (n = 13) | 7–14 (n = 11) | −7 to 1 | 9–14 (n = 11) | −5 to −3 | 5–14 (n = 8) | −7 to −1 | 7–12 (n = 7) | −4 to 1 |
| L3-40 | 10 | 10 | −1 | 12 | +1 | 12 | +2 | 8 | −1 |
| | 9–13 (n = 11) | 9–15 (n = 11) | −3 to 2 | 9–18 (n = 10) | −4 to 6 | 10–17 (n = 8) | −1 to 3 | 6–22 (n = 9) | −5 to 8 |

Bold data indicate the presence of significantly different values.
ᵃSignificantly different from baseline.
ᵇSignificantly different from Placebo.

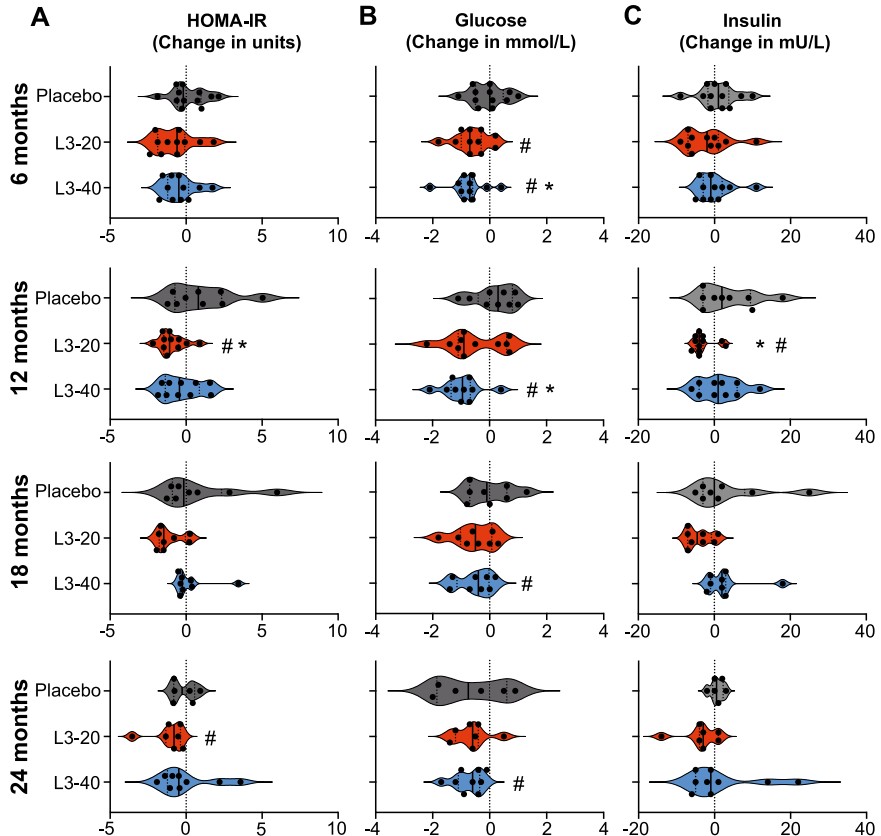

**Fig. 3 | Effect of experimental hookworm infection on insulin resistance (HOMA-IR), fasting blood glucose and insulin.** Changes in the (**A**) Homoeostatic model assessment of insulin resistance (HOMA-IR, units), **B** fasting blood glucose and (**C**) fasting blood insulin levels from baseline values at each 6-monthly evaluation visit. Violin plots display each individual data point, median and IQR. Dotted vertical line indicates baseline levels (0 change). *Significant difference to placebo, Kruskal–Wallis test with Dunn's multiple comparisons. #Median value significant difference to baseline, Durbin, Skillings–Mack test and Wilcoxon signed-rank test. Source data are provided as a Source Data File.

maximum of 12 [10–15] at 6 months (Table 4 and Fig. 3C). There were no longitudinal changes in insulin levels in the L3-40 group at any time point and no significant differences compared to Placebo. Insulin levels in the L3-20 group were consistently lowered from the baseline level of 13 [12–16] to 10 [9–14] at 12 months ($p = 0.031$ compared to baseline), with reductions from baseline values reaching statistical significance compared to Placebo at 12 months (−4.0 in L3-20 compared to +2.0 in Placebo; $p = 0.02$).

Levels of glycated haemoglobin (HbA1cIFCC, mmol/mol) displayed an upward trend at all evaluation visits compared to baseline in the Placebo group, ranging from a baseline value of 32 [30–33] mmol/mol to a maximum of 35 [33–37] at 2 years (Supplementary Table 2). Values in the L3-20 group were stable near the baseline value of 33 [30–35] throughout, while values in the L3-40 group experienced similar upward trends to the Placebo group, with values significantly higher than baseline at 18 months ($p = 0.031$).

### Secondary metabolic outcomes: body mass and blood lipid profile

Median body mass and Body Mass Index (BMI) in the Placebo group remained stable throughout (Table 5, Fig. 4 and Supplementary Fig. 5). While there were no statistically significant differences in body mass or BMI in the L3-20 or L3-40 groups compared to Placebo (Table 5), body mass and BMI in the L3-20 group tended to be lower than their baseline values at all evaluation visits, with significant reductions in body mass and BMI in the L3-20 group at 18 months ($p = 0.031$ for body mass, $p = 0.016$ for BMI) and 24 months ($p = 0.031$ for body mass and $p = 0.016$ BMI). This corresponded to a median reduction in body mass

of approximately 5 kg at both time points in the L3-20 group (Table 5 and Fig. 4) and a reduction in BMI of 2.2 units and 2.4 units at 18 months and 2 years, respectively (Table 5 and Supplementary Fig. 5). No significant longitudinal changes in body mass or BMI were evident in the L3-40 group.

Analysis of blood lipid profile revealed that median levels of total cholesterol, triglycerides, high-density lipoprotein, low-density lipoprotein and total cholesterol/HDL ratio remained relatively stable in the Placebo group at all time points, with only a statistically significant reduction in LDL at 12 months ($p = 0.016$) and 18 months ($p = 0.023$) (Supplementary Tables 3 and 4). In both the L3-20 and L3-40 groups, there were no longitudinal changes in any of these parameters compared to baseline and no significantly different changes compared to Placebo other than elevated LDL ($p = 0.019$) and total cholesterol ($p = 0.038$) at 18 months in the L3-40 group compared to Placebo (Supplementary Tables 3 and 4)

### Discussion

Over recent decades, helminth infection has emerged as a potential approach for treating allergic and autoimmune disorders, as well as metabolic disorders such as Metabolic Syndrome and type 2 diabetes[15,21,27]. Until now, a potential supportive role for worms in metabolic health has relied on animal studies and human cross-sectional or deworming studies that could not infer causality. The present study provides human clinical trial evidence supporting the potential beneficial effect of helminth infection in metabolic disease. Infection with low doses of hookworms caused significantly reduced measures of glucose homoeostasis (lowered HOMA-IR and fasting

**Table 5 | Body mass and Body Mass Index (BMI) absolute values and changes in each treatment group**

| | baseline | 6 months | Δ6 mo. | 12 months | Δ12 mo. | 18 months | Δ18 mo. | 24 months | Δ24 mo. |
|---|---|---|---|---|---|---|---|---|---|
| *Body mass (kg), Median and IQR (n)* | | | | | | | | | |
| Placebo | 105 | 107 | +2 | 99 | 0 | 105 | −1 | 106 | +2 |
| | 89–115 (n = 12) | 91–115 (n = 12) | −4 to 4 | 90–113 (n = 8) | −4 to 4 | 89–119 (n = 8) | −8 to 9 | 91–118 (n = 6) | −4 to 5 |
| L3-20 | 103 | 101 | −3 | 104 | −1 | **93**[a] | −5 | **93**[a] | −5 |
| | 86–131 (n = 13) | 82–134 (n = 12) | −3 to 1 | 90–133 (n = 11) | −5 to 1 | 76–120 (n = 8) | −12 to −3 | 77–124 (n = 7) | −10 to −1 |
| L3-40 | 102 | 102 | −1 | 106 | +1 | 109 | +2 | 108 | +4 |
| | 90–131 (n = 11) | 86–130 (n = 11) | −2 to 1 | 91–129 (n = 10) | −2 to 6 | 100–136 (n = 8) | −1 to 8 | 94–133 (n = 9) | −2 to 6 |
| *BMI (kg/m$^2$), Median and IQR (n)* | | | | | | | | | |
| Placebo | 37 | 37 | +0.7 | 36 | +0.1 | 39 | −0.3 | 38 | +0.8 |
| | 33–40 (n = 12) | 33–41 (n = 12) | −1.2 to 1.2 | 31–41 (n = 8) | −1.5 to 1.5 | 32–41 (n = 8) | −3.0 to 3.2 | 32–40 (n = 6) | −1.9 to 1.7 |
| L3-20 | 37 | 37 | −0.5 | 39 | −0.2 | **35**[a] | −2.2 | **33**[a] | −2.4 |
| | 33–46 (n = 13) | 32–49 (n = 12) | −1.3 to 1.5 | 34–44 (n = 11) | −1.3 to 0.1 | 31–41 (n = 8) | −4.2 to −1 | 30–41 (n = 7) | −2.7 to −0.5 |
| L3-40 | 36 | 36 | −0.1 | 35 | +0.3 | 39 | +0.6 | 38 | +0.9 |
| | 33–40 (n = 11) | 31–40 (n = 11) | −0.7 to 0.5 | 33–39 (n = 10) | −0.8 to 2.3 | 34–40 (n = 8) | −0.2 to 2.3 | 33–41 (n = 9) | −0.7 to 2.1 |

Bold data indicate the presence of significantly different values.
[a]Significantly different from baseline.

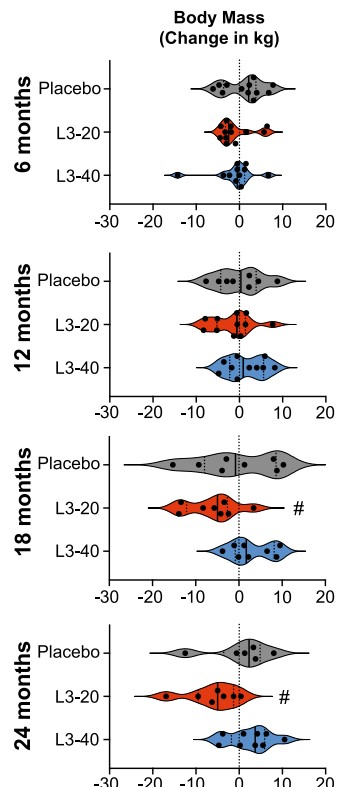

**Fig. 4 | Impact of experimental hookworm infection on body mass.** Changes in participant's body mass (kg) from baseline values at each 6-monthly evaluation visit. Violin plots display each individual data point, median and IQR. Dotted vertical line indicates baseline levels (0 change). #Median value significant difference to baseline, Durbin, Skillings-Mack test and Wilcoxon signed-rank test: Source data are provided as a Source Data File.

blood glucose) compared to pre-infection and Placebo treatment, which was associated with modest body mass reductions. Regular monitoring of diet and physical activity habits revealed no consistent changes that could have contributed alone to improved insulin resistance. While the study was relatively small and needs to be validated by larger follow-up studies, the results provide proof of principle that hookworms and/or the biological changes they invoke in their human hosts are a safe and potentially beneficial intervention for improving determinants of metabolic health.

The primary outcome of this Phase Ib study was to determine the safety and tolerability of hookworm infection in people at risk of developing T2D. While hookworm infection was associated with a higher incidence of gastrointestinal AEs, these were anticipated during the time that hookworms established in the gut[30] and were typically mild-moderate and resolving. Three of the 28 hookworm-treated participants displayed GI symptoms that warranted deworming, and symptoms resolved promptly in two. The third did not improve after deworming, suggesting their symptoms were unrelated to hookworm infection. Overall, the safety profile of hookworm treatment in people at risk of T2D is similar to that in other diseases[30].

In the current study, hookworm infection was associated with improved insulin resistance, consistent with previous mouse or natural human helminth infection studies[7,21,32]. Lowered HOMA-IR was most apparent in the group that received 20 hookworms and was most pronounced at 18 months, with a reduction in median HOMA-IR from 3.0 units at baseline (consistent with pre-diabetic insulin resistance) to 1.6 units at 18 months (a healthy level). The large Cohen's *d* effect sizes in the L3-20 group highlight the statistical and clinical significance of this sustained improvement in metabolic health. Interestingly, the improvements in HOMA-IR in the group that received 40 worms were less dramatic, despite similar trends in reduced HOMA-IR at 6 and 12 months. Both the L3-20 and L3-40 groups showed similar reductions in fasting blood glucose compared to Placebo throughout the study, but only the L3-20 group displayed sustained reductions in insulin. By the later stages of the study, all results from the L3-40 group were comparable to Placebo. Notably, we did not detect consistently higher degrees of hookworm infection intensity (hookworm EPGs) or significantly stronger eosinophil responses with the higher dose of worms, and correlation analyses did not show a significant association between higher worm doses and faecal EPGs with greater improvements in HOMA-IR. Clearly, larger clinical trials are required to clarify the optimal dose of hookworms. One factor that may partially explain why participants in the L3-20 cohort responded better to treatment was that pre-trial median age and HOMA-IR were higher; hence there was greater potential for improvements. Future clinical trials designed to demonstrate efficacy should stratify participants based on high and low baseline HOMA-IR.

In conversations with potential participants during recruitment and screening, it was evident that a powerful motivator to enrol was

the prospect of body mass reductions, as documented in mouse[20,22,24] or natural human helminth infection studies[7,24,32,33]. We noted modest reductions in body mass and BMI in the L3-20 group; however, body mass reductions were not consistent for all hookworm-treated participants, possibly due to the relatively low worm burdens compared to those seen in animal studies or natural infections. This caused disappointment for some people, who then elected to pursue gastric sleeve surgery despite improvements in HOMA-IR. Most participants were less concerned and continued with unabated enthusiasm. Bar one, all hookworm-treated participants that completed the study opted to retain their worms, and the sole person who dewormed did so in preparation for a medical procedure.

Interestingly, hookworm infection did not show significant improvements in blood lipid profiles, in contrast to what may have been expected based on previous studies[9,10,13,24,33]. A potential explanation could lie in the different helminth species used in the current trial. Previous studies noting improved blood lipid markers included cross-sectional and deworming studies of infections with multiple soil-transmitted helminths[33] and previous and current schistosomiasis[9,10,13,24] but not hookworm infection alone. Results from a 2019 systematic review and meta-analysis suggested that different helminth species may have different efficacy in altering lipid and glucose homoeostasis[15]. The authors postulated that the liver-specific residence of Schistosoma spp. compared to intestinal helminths could facilitate greater metabolic improvements in schistosomiasis. Future helminth challenge studies such as ours that instead use single-sex infections with schistosome parasites[34] may validate this hypothesis.

The study had several limitations, with the major shortcoming being the small sample sizes that limited our ability to make firm conclusions. This was further curbed by the long duration of the study and attrition of participants in all groups after the 12-month time point. Since T2D takes many years to manifest, future studies will need to follow up with participants for longer periods. Ideally, this would be combined with repeated doses of hookworms to maintain a chronic infection and monitoring of the associated immune response. Such studies could include a deworming arm to determine if an active hookworm infection is required for longer-term benefits and could use HbA1c IFCC as a longer-term primary outcome measure of improved glycaemic control. Additionally, participant selection bias limited the generalisability of our findings. Most participants were of Caucasian descent, and it is unclear if different populations, such as people of Asian descent or Australian First Nations people, who disproportionately suffer from T2D[35], would experience similar improvements. Further, our qPCR methods for quantifying hookworm infection intensity (EPGs) may not accurately reflect the true level of hookworm burden. Our qPCR assay was based on the quantification of non-embryonated eggs, and in our study, faecal samples provided by participants were not always immediately frozen, potentially allowing some eggs to embryonate. This would cause the levels of parasite DNA to increase, resulting in the qPCR overestimating parasite egg burdens. Future studies should use ethanol fixation, which would allow for better standardisation of methods for determining hookworm infection intensities. Lastly, within the scope of this study, we could not attain a mechanistic understanding of how and why hookworm infection caused improvements in insulin resistance. It remains conceivable that improvements in insulin resistance could have been a direct consequence of reduced body mass after hookworm treatment; however significant reductions in body mass were not observed until after significant improvements in HOMA-IR were seen. Nevertheless, future studies that are designed to demonstrate efficacy should attempt to explore the potential kinetic relationships between hookworm-mediated body mass reductions and changes in insulin resistance. Regarding potential direct mechanisms of hookworm-mediated improvements in metabolic health; type 2 immune responses and eosinophils are known to counteract detrimental pro-inflammatory cytokine responses and positively influence metabolism[20,36], and we did observe consistent elevations in circulating eosinophils and type 2 cytokines such as IL-5, so it remains plausible that they contributed to stimulating this pathway. However, we did not observe any evidence that hookworm infection supressed systemic pro-inflammatory cytokine levels other than modest and transient reductions in IL-17A. It should be noted however that the time points at which we assessed changes in immune profiles (6 and 12 months) were likely too late to detect the more acute changes that would have occurred as the worms established in the human host[11]. Better understanding of the beneficial biological effects of hookworms in the context of metabolic syndrome may identify novel therapeutic strategies that could mimic the actions of the live hookworm (e.g. immune-modifying or microbiome therapies) or may pinpoint factors that are excreted/secreted by hookworms that could be produced as novel biologic therapies[37].

In conclusion, the hygiene hypothesis[8,12] provides a persuasive explanation for the increasing prevalence of inflammatory and metabolic disorders in populations with a low prevalence of helminth infections. The present study suggests that experimental infection with low hookworm doses is safe and is associated with improvements in glucose homoeostasis in people with Metabolic Syndrome and at risk of type 2 diabetes mellitus. Results from this proof of concept study will inform the further development of novel preventative interventions in humans at risk of type 2 diabetes.

## Methods

### Study design and regulatory approvals
Between January 2018 and April 2022, we conducted a 2-year Phase Ib randomised, double-blinded, placebo-controlled trial of experimental hookworm infection (either 20 or 40 infective third-stage larvae, L3) in otherwise healthy people at risk of T2D. The study included two clinical sites in Queensland, Australia (James Cook University Cairns and Townsville campuses). The study designated C26 was approved on April 19th 2017 by the Human Research Ethics Committee of James Cook University. Recruitment progressed from January 2018 until June 2020 and data was collected until April 2022. The conduct of this clinical trial research study complied with all relevant ethical regulations. All participants gave written and informed consent to be part of the study. The trial was registered with the Australian New Zealand Clinical Trials Registry (ACTRN12617000818336). The clinical study protocol was published in 2019[38].

### Participants and sample collection
Inclusion criteria: Otherwise healthy males and females aged 18–50 years with central obesity (waist circumference> 90 cm for women and >102 cm for men) and increased insulin resistance as assessed via abnormal HOMA-IR > 2.12 or at least two other features of metabolic syndrome determined at screening (blood pressure >135/85 mmHg, dyslipidaemia or abnormal liver function tests). Exclusion criteria, in brief, included pregnancy, established chronic disease, historical or current substance abuse, major allergies, known immunodeficiency disorder, unstable asthma or taking medications likely to interfere with study outcomes. Blood samples were taken at each 6-monthly visit into serum separator tubes for serum analyses or heparin-coated tubes for complete blood count analysis (Beckton Dickinson). Participants were asked to provide fresh faecal samples (within 24 h of collection) at each study visit, which were then frozen at −80 °C for future analysis of the presence of hookworm eggs by qPCR. Participants were not compensated for their participation.

## Sample size calculation, randomisation and masking

In the absence of other effect size data from human trials upon which to base a power calculation, and since a change in HOMA-IR from baseline was the key metabolic outcome, we adapted the SUGARSPIN HOMA-IR result[39] and assumed an effect size of 1.06 over 2 years. A total of 15 participants in each group reflected 80% power to detect an effect size of 1.06 using the T-statistic and 1.023 using Z-statistic. An initial recruitment target of 54 participants allowed for a dropout rate of 20%, leaving a desired recruitment target of 45 volunteers (15 in each treatment group) to participate in the trial. Volunteers who met the eligibility criteria and had given informed consent were block randomised (block size of 6) on a 2:2:2 ratio according to a computer-generated sequence to one of the three study arms: Placebo, 20x *N. americanus* L3 (L3-20) or 40x L3 (L3-40). Participants and investigators were blinded to the treatment, other than the assigned producer of the inocula (co-author L.B.).

## Preparation and administration of treatments

The inocula were prepared freshly as previously described[40]. Hookworm ova were collected from two volunteer donors, initially infected in 2013 or 2018 with 5*N. americanus* L3 from a line donated by Professor David Pritchard (University of Nottingham). Visibly motile L3 were individually selected for inclusion in the inocula, contained within 300 μl of de-ionised water. Placebo inocula comprised 300 μl of de-ionised water containing approximately 2 μl of Tabasco® sauce. The inoculation procedure involved dispensing the solution onto a non-absorbent dressing pad placed onto the participant's forearm. Two doses of inocula were administered eight weeks apart, with L3-20 and L3-40 participants receiving 10x or 20x L3 on each occasion (Fig. 1A). Hookworm infection status was monitored in 1 g of thawed and homogenised faecal samples collected at baseline and six months using qPCR and formulas as described previously[41,42].

## Outcome measures

Safety and general health assessments and were conducted at each designated visit (Fig. 1A) to identify adverse events (AEs) and serious AEs and their suspected causality. Complete blood counts and metabolic indices (glucose, insulin, HbA1c, liver function tests and lipids) were measured in serum samples at an accredited Australian clinical pathology laboratory. Cytokines were measured in thawed serum samples using a multiplex human cytokine bead array panel according to manufacturers instructions (BioLegend LEGENDplex, catalogue #741027). HOMA-IR was calculated by multiplying fasting blood glucose (mmol/L) by insulin (mU/L) values and dividing by 22.5[29]. Blood pressure was monitored using automated blood pressure devices. Height was measured using either a wall-mounted measuring tape or a stadiometer. Waist circumference was measured using a flexible measuring tape. Body mass was recorded using digital scales with a maximum capacity of 250 kg. Body mass index (BMI) was calculated by dividing mass in kg by height in metres squared. Additional exploratory secondary outcomes outlined in our study protocol[38] such as microbiome and polyunsaturated fatty acid profile were beyond the scope of this clinical report and will be published elsewhere.

## Diet, physical activity, mood and depressive state monitoring

The 14-item self-administered Prevención con Dieta Mediterránea (PREDIMED) diet questionnaire was used to monitor changes in participants' diets throughout the trial[43]. Using a scorecard, participants recorded their weekly physical activity, including type, duration, frequency, and intensity, which were combined into a single metric[44], the metabolic equivalent of task (MET) per week (MET × duration × frequency = METs/week). To track changes in mood and depressive state, we used the validated Patient Health Questionnaire (PHQ)-9[45].

## Statistical analyses

The chi-square test for trend was used to determine differences in the safety of hookworm infection, assessed by the proportions of AEs between each group. The Kaplan–Meier model was used to compare progression through the study, with associations between groups assessed using the Log-rank test statistic. Standard descriptive statistics were performed on all characteristics at baseline and each of the four evaluation visits (categorical variables: absolute and relative frequencies; numerical variables: mean and 95% confidence intervals or median and interquartile range, dependent on data distribution). Normality testing indicated that most data were not normally distributed. Thus, Kruskal-Wallis with Dunn's multiple comparisons tests were used to detect inter-cohort differences in absolute values and changes. Mann–Whitney tests (PHQ-9) or Friedman's test with Dunn's multiple comparisons (cytokines) were used when hookworm groups were combined. For significant results in tests with posthoc analyses, the adjusted *p*-value is reported. For test results stated as not significant, *p* was > 0.05.

As suggested in Tomczak and Tomczak[46], eta-squared ($\eta^2$) was calculated for the Kruskal–Wallis test as a non-parametric measure of overall effect size ($\eta^2 = H-k + 1/n-k$). Eta-squared can then be transformed into $d_{Cohen}$ for easy comparison using an online calculator (https://www.psychometrica.de/effect_size.html#transform). Similarly, the *Z* test statistic from Dunn's multiple comparisons tests was used to compute the value of the correlation coefficient *r* as an effect size estimate for the pairwise comparisons at each time point ($r = Z/\sqrt{n}$); the correlation coefficient *r* was further transformed into $d_{Cohen}$ using the formula $d = 2r/\sqrt{(1-r^2)}$[46] and results confirmed via the online calculator.

A research team member and an independent monitor reviewed data from each participant to identify and address missing information. To account for missing data, this study followed the method prescribed in Jakobsen et al.[47]. Specifically, as missing data exceeded 40%, no imputation was undertaken, and results were interpreted in the context of this limitation. The non-parametric Durbin, Skillings–Mack test for incomplete block design, which allows missing data, was used for longitudinal analysis, followed by pairwise multiple comparison by Wilcoxon signed-rank test with Holm–Bonferroni correction to identify significant differences between baseline and each evaluation visit. Statistical analyses were performed in GraphPad Prism 9.0 and XLSTAT 2023 for Excel.

## Role of the funding source

The funders had no role in the design or conduct of the study.

## Reporting summary

Further information on research design is available in the Nature Portfolio Reporting Summary linked to this article.

## Data availability

All data that support the findings of this study, including deidentified individual data, are available within the paper and its Supplementary Information files. All data are available immediately following publication to anyone who wishes to access the data, with no end date, and for any purpose. The clinical study protocol is already published[38], which contains additional study information such as the Informed Consent form. Source data are provided with this paper.

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

## Acknowledgements
The work was funded by the Far North QLD Hospital Foundation (P.G., R.M., A.L.), Australian Institute of Tropical Health and Medicine (R.M., P.G., A.L.), Australian National Health and Medical research Council (NHMRC) Senior Principal Research Fellowship (A.L.), Program Grant (A.L.), Advance Queensland Fellowship (P.G.) and an Australian Research Training Program Stipend (D.P.). Funders had no role in the design or conduct of the study. We also thank Sally McDonald, Lynne Reid, Melissa Piontek, Tyler Gilstrom, Melissa Campbell and Geraldine Buitrago for assistance with clinical trial operations.

## Author contributions
Conceptualization: R.M., A.L., T.R., J.C., and P.G. Data analysis: D.P., F.T., R.M., P.G. Funding: D.P., A.L., R.M., P.G., M.F. Investigation: D.P., M.M., L.M., L.B., C.L., A.L., R.M., P.G. Methodology: D.P., M.M., L.B., J.C., A.L., R.R., P.Z., S.F., R.T., R.M., P.G., M.F. Project administration: D.P., M.M., L.M., F.T., C.L., R.M., P.G. Writing—original draft. D.P. and P.G. Writing—review and editing: all authors.

## Competing interests
P.R.G and A.L, are founders and shareholders of Macrobiome Therapeutics, which is developing hookworm-derived proteins as drugs for treating inflammatory conditions. The remaining authors declare no competing interests.
