## [Peer Review File · Nature Communications]

Effect of experimental hookworm infection on insulin resistance in people at risk of type 2 DiabetesREVIEWER COMMENTS

Reviewer #1 (Remarks to the Author):

This is a first in human (phase 1b) trial on the effect of hookworm infection on insulin resistance in people at risk of Type 2 diabetes. Primary outcome is safety, intervention groups showed higher numbers of AEs, but these were as anticipated, and safety profile was considered similar to that in other diseases. Due to limitations of small sample size, high drop-out and unbalance in baseline age and HOMA-IR, it is difficult to compare metabolic outcomes and it should be made clear that the results are to be considered with caution and as hypothesis generating. Not too much weight should be given to significant comparisons and non-significant results should never be interpreted as absence of effect. The data suggest anyway some signal of an effect that could encourage further research.

1. Sample size calculation/justification has not been described in the methods (and I am not sure why methods section is at the end of the paper)
2. Has any blocking (block size) been used in the randomisation?
3. All analyses used non-parametric methods because data were not normally distributed but the effect size are calculated using Cohen's d. For consistency a non-parametric effect size should be reported.
4. It is stated that both ITT and PP analyses were performed. Which one was the primary analysis? (although considering the high number of missing values and no missing imputation, it cannot be considered as an actual ITT analysis).
5. Some more details would be needed regarding the longitudinal analyses. It is reported that for longitudinal analyses mixed effects model (is it linear mixed effect model?) was used, and this would compare means but medians are presented. Are the comparisons between single time periods estimated also from the mixed model? Was time*treatment group interaction included in the model?
6. In the discussion some results are a bit over-interpreted with comments like "the first reported controlled-trial clinical evidence for the beneficial effect of helminth infection in metabolic disease" as the results should be interpreted with caution and this is not a definitive trial, or "Diet and physical activity changes could not explain these improvements" as the study is not power to answer this question.

Reviewer #2 (Remarks to the Author):

Pierce et.al. report a clinical trial of experimental hookworm infection in patients at risk of Type 2 diabetes. This randomized placebo controlled trial is an impressive accomplishment, although ultimately data is only available for approximately 7-9 people per group (hence the number 40 listed in the abstract is misleading). Overall, the data supports the hypothesis that hookworm infection may have beneficial metabolic effects on infected people.

However, there are some important questions that should be addressed. Most importantly,

it is not reported the level of infection of the study participants (just PCR positive or negative). Egg counts or some other measure (e.g. quantitative PCR) of infection intensity could be very important because there is substantial heterogeneity in establishment of infection. The variation in metabolic effects may be related to differences in colonization efficiency between the different people.

Also, it is unclear if the worms are prepared and delivered in different “batches” which can also be variable in terms of infectivity. I’m not clear if the people are randomized to the 20 vs 40 L3 dose, so maybe some “good” batches all went into the people who got 20 L3’s hence a stronger effect than 40. However, this can also be examined through the egg counts. Because the people that got 40 L3’s should obviously have higher egg counts than the people who got 20, unless there is a problem with the worm preparations. Did higher worm dose actually lead to higher worm colonization? This is unclear.

The investigators should also examine the relationship between the adverse events and egg counts/worm burdens. Are the people who develop moderate symptoms the individuals with higher egg counts? Hence, it may be possible to tell if adverse events is driven by the host response, or the parasite burden.

Apart from CBC w differential data, not much information is provided on the immune response of the study participants, it would be interesting to know if any changes in Th1/Th2 or Treg responses occur during infection longitudinally.

Overall, this is a novel and important pilot study to investigate if hookworm infection affects human metabolism in a similar fashion to what has been reported in mouse models. It raises the possibility of conducting larger clinical studies for establishing efficacy in larger numbers of participants.

Reviewer #3 (Remarks to the Author):

The reduced prevalence of insulin resistance and type 2 diabetes (T2D) in countries with endemic parasitic worm infections suggests a protective role for worms against metabolic disorders, however clinical evidence has been non-existent. This study investigated this hypothesis.

Overall comments:

This is a fun and novel approach. I like it.

However, I do have some concerns which should be further discussed.

Since exercise and energy intake also have pronounced effect of insulin resistance, could this affect the results?

See eg:

Healthy weight loss maintenance with exercise, liraglutide, or both combined
JR Lundgren, C Janus, SBK Jensen, CR Juhl, LM Olsen, RM Christensen, ...
New England Journal of Medicine 384 (18), 1719-1730

and

Exploratory analysis of eating-and physical activity-related outcomes from a randomized
controlled trial for weight loss maintenance with exercise and liraglutide single or ...
SBK Jensen, C Janus, JR Lundgren, CR Juhl, RM Sandsdal, LM Olsen, ...
Nature Communications 13 (1), 4770

Did the authors consider the health care systems and the level of testing for T2D in countries
with worm infections?

Please see eg. Health care professionals from developing countries report educational
benefits after an online diabetes course

NJ Wewer Albrechtsen, KW Poulsen, LØ Svensson, L Jensen, JJ Holst, ...
BMC medical education 17, 1-8

Did the authors consider the minimum relevant effect size between groups?

Pierce et al. “Effect of experimental hookworm infection on insulin resistance in people at risk of Type 2 Diabetes”

We thank the three reviewers of our manuscript for their positive comments and helpful suggestions for improving our manuscript. Below is our response to the Reviewer’s comments.

Reviewer 1:

This is a first in human (phase 1b) trial on the effect of hookworm infection on insulin resistance in people at risk of Type 2 diabetes. Primary outcome is safety, intervention groups showed higher numbers of AEs, but these were as anticipated, and safety profile was considered similar to that in other diseases. Due to limitations of small sample size, high drop-out and unbalance in baseline age and HOMA-IR, it is difficult to compare metabolic outcomes and it should be made clear that the results are to be considered with caution and as hypothesis generating. Not too much weight should be given to significant comparisons and non-significant results should never be interpreted as absence of effect. The data suggest anyway some signal of an effect that could encourage further research.

1. Sample size calculation/justification has not been described in the methods (and I am not sure why methods section is at the end of the paper)

Response: We agree that this information, which was originally only in our published study protocol, should also be included in this manuscript. Please see **Page 24, lines 471-476** for the sample size determination information.

The Methods section is provided at the end of the paper, as per the Journal’s formatting requirements.

2. Has any blocking (block size) been used in the randomisation?

Response: We apologise for the omission of this information in the original submission; this information has now been included in the revised Methods section (**Page 24, lines 479-480**).

3. All analyses used non-parametric methods because data were not normally distributed but the effect size are calculated using Cohen’s d. For consistency a non-parametric effect size should be reported.

Response: We thank the reviewer for highlighting this important issue. In our original submission, we had erroneously calculated Cohen’s d based on mean values and have now corrected this, using a method that uses statistics from non-parametric tests to calculate effect sizes.

Details of the new methodology are included in the revised Methods section (**Page 26, lines 536-543**) and the corrected effect sizes are now reflected in the updated results section (**Pages 11/12, lines 239-245**) and revised **Supplementary Table 1**.

4. It is stated that both ITT and PP analyses were performed. Which one was the primary analysis? (although considering the high number of missing values and no missing imputation, it cannot be considered as an actual ITT analysis).

Response: The primary outcome analysis was an Intention to Treat analysis of safety, determined by incidence of Adverse Events. We have clarified this in the revised results sections on **Page 7, line 125**. We have also clarified that secondary endpoints were assessed using a Per Protocol analysis (**Page 8, line 152**).

5. Some more details would be needed regarding the longitudinal analyses. It is reported that for longitudinal analyses mixed effects model (is it linear mixed effect model?) was used, and this would compare means but medians are presented. Are the comparisons between single time periods estimated also from the mixed model? Was time*treatment group interaction included in the model?

Response: We thank the reviewer for raising these questions. The linear mixed effects model was originally used for longitudinal analyses due to missing values, which prohibited the application of Friedman's test for repeated measures. No alternative non-parametric test was available in the statistical software we originally used (GraphPad Prism).

We have now sourced an alternative software that allowed the use of a non-parametric repeated measures test in an incomplete block design (XLSTAT for Excel). The details are now included in the updated Methods section (Page 27, line 548-553), and all relevant tables, figures and results section text have been corrected accordingly with the appropriate *p* values. Time*Treatment Group Interaction was not included in the model and was not the focus of the planned data analyses.

6. In the discussion some results are a bit over-interpreted with comments like “the first reported controlled-trial clinical evidence for the beneficial effect of helminth infection in metabolic disease” as the results should be interpreted with caution and this is not a definitive trial, or “Diet and physical activity changes could not explain these improvements” as the study is not power to answer this question.

Response: We agree that it would be appropriate throughout the manuscript to recognise the limitations of the study's sample size, and be careful to not over-interpret the data. Please see highlighted text on Page 22, lines 441-443 and Page 18, lines 334-336 and 330-331 in the revised discussion where we have changed wording in this regard.

Reviewer 2:

Pierce et.al. report a clinical trial of experimental hookworm infection in patients at risk of Type 2 diabetes. This randomized placebo controlled trial is an impressive accomplishment, although ultimately data is only available for approximately 7-9 people per group (hence the number 40 listed in the abstract is misleading). Overall, the data supports the hypothesis that hookworm infection may have beneficial metabolic effects on infected people.

However, there are some important questions that should be addressed. Most importantly, it is not reported the level of infection of the study participants (just PCR positive or negative). Egg counts or some other measure (e.g. quantitative PCR) of infection intensity could be very important because there is substantial heterogeneity in establishment of infection. The variation in metabolic effects may be related to differences in colonization efficiency between the different people.

Response: We agree with the reviewer and we have now undertaken more quantitative tests of hookworm infection intensity (faecal *N. americanus* eggs per gram, EPG) in study participants where samples were available for analysis at 6 months post-infection. Methodological details are described in the revised submission (Page 24, lines 494-496). A new panel in Figure 2 has now been added to the manuscript (Figure 2D) and is copied to the right, which as the reviewer predicted, shows substantial heterogeneity in establishment of infection between individuals (medians, with all data points shown). This finding has been discussed in the relevant section of the results text (Page 10, lines 202-205).

To address whether this variation may have contributed to differences in metabolic effects, we have correlated infection intensity (EPGs) with changes in insulin resistance (HOMA-IR) in hookworm treated participants at week 26, where data for both parameters were available. As can be seen below, there is no clear association between infection intensity and improvements in insulin resistance- the linear regression analysis (solid line) is a near horizontal line. We think that this new data is informative however, so we have included this analysis as new supplementary data (**Supplementary Figure 5**) and have interpreted these findings in a revised results and discussion (**Page 11, lines 233-237 and Page 19, lines 362-366**), which concludes that- at least in the doses we have tested in this trial- that more hookworms does not necessarily correlate with greater improvements in insulin resistance.

Also, it is unclear if the worms are prepared and delivered in different “batches” which can also be variable in terms of infectivity. I’m not clear if the people are randomized to the 20 vs 40 L3 dose, so maybe some “good” batches all went into the people who got 20 L3’s hence a stronger effect than 40.

Response: This is an important point raised by the reviewer, which required us to add extra information to clarify. Participants were block randomised to received placebo, L3-20 or L3-40 treatment in groups of 6 (2:2:2), so while multiple batches of worms were indeed produced for this study, people from the L3-20 group typically received the same batch of larvae as people from the L3-40 group, when randomised to treatment within the same week. We have included the detail about block randomisation in the Methods (**Page 24, lines 479-480**).

However, this can also be examined through the egg counts. Because the people that got 40 L3’s should obviously have higher egg counts than the people who got 20, unless there is a problem with the worm preparations. Did higher worm dose actually lead to higher worm colonization? This is unclear.

Response: This point raised by the reviewer is closely related to the one above regarding quantifying infection intensity. The data in new **Figure 2D** shows that while the median EPG in the L3-40 group tends to be higher than the L3-20 group, as would be expected, there is no clear statistical difference due to high levels of inter-individual variability.

However, there are limitations with our ability to interpret these quantitative findings. Due to the double blinded nature of the study we were unable to enumerate parasite eggs in stool samples using standard microscopy in real-time, hence we need to rely on retrospective qPCR quantification. The qPCR methodology and formulas that we used to quantify EPGs were based on the presence of non-embryonated eggs. In our study, faecal samples that were provided to us by participants were not always immediately frozen or preserved in a fixative, they were often refrigerated for hours or overnight before delivery. If embryonation of the eggs occurs, the levels of parasite DNA increases- hence affecting the qPCR determination. Hence there is a possibility that some of the variability we observed may have been due to variations in the timing of when samples were frozen. These limitations are described in the revised discussion (**Page 21, lines 409-416**).

The investigators should also examine the relationship between the adverse events and egg counts/worm burdens. Are the people who develop moderate symptoms the individuals with higher egg counts? Hence, it may be possible to tell if adverse events is driven by the host response, or the parasite burden.

Response: We thank the reviewer for this comment. We have only a limited set of data to address this, but we have attempted to address this question by comparing the infection intensity (EPGs) in hookworm-treated participants who reported a gastrointestinal (GI) adverse event, versus people who reported no GI adverse events. As you can see by the graph below (medians, with all data points shown), there is a very weak, non-statistically significant trend where people who reported GI symptoms had higher median faecal hookworm EPGs than those who reported no GI symptoms. However the variability in the data is high, and it is apparent that two of the people who reported no GI symptoms had the second and third highest EPGs recorded in the study- so it is not appropriate to say that higher infection intensity was associated with greater adverse symptoms.

We think that these data are not conclusive enough to provide in the revised manuscript; and have provided them here as “reviewer only” data. We leave it up to the discretion of the Editor and Reviewer if you feel that it should be included as a supplementary figure.

Apart from CBC w differential data, not much information is provided on the immune response of the study participants, it would be interesting to know if any changes in Th1/Th2 or Treg responses occur during infection longitudinally.

Response: We agree with the reviewer that some immune profiling beyond the already included CBC counts would add value to the manuscript. Consequently, we have undertaken a multiplex cytokine analysis of serum samples collected at baseline, 6 months and 12 months post-hookworm infection, to assess whether the hookworm treatments were inducing Type-1, Type 2 or regulatory immune responses in people at risk of Type 2 diabetes. Results from both hookworm cohorts L3-20 and L3-40 were combined because there was no clear quantitative differences between responses with the different worm doses.

Data that we are including in a new supplemental figure (**Supplementary Figure 3**) and copied below show that, as expected, hookworm treatment was associated with induction of a biased Type 2 immune response, particularly the significantly increased levels of IL-5 at 6 and 12 months. Data are presented as median cytokine levels (+/- IQR), and were compared using a Friedman’s test with Dunn’s multiple comparisons. Levels of other Type-2 associated cytokines such as IL-4 and IL-13 tended to be elevated above baseline, but did not reach statistical significance due to large degrees in variability between participants. There were no significant alterations in the levels of pro-inflammatory Type-1 response cytokines such as IFN-gamma, TNF, or IL-2. IL-17A levels were significantly reduced at 6 months after treatment, but levels of regulatory cytokines such as IL-10 were not significantly elevated.

Altogether, these data complement the already included CBC data, highlighting that immune responses to hookworms in people at risk of Type 2 diabetes were as anticipated based on previous human studies (e.g. Eosinophilia, elevated IL-5). These findings are discussed in the relevant section of the Results (Page 10, lines 202-211) and Discussion (Page 21, lines 423-433), and methodological information is included in the revised Methods (Page 25, lines 502-504).

Overall, this is a novel and important pilot study to investigate if hookworm infection affects human metabolism in a similar fashion to what has been reported in mouse models. It raises the possibility of conducting larger clinical studies for establishing efficacy in larger numbers of participants.

Response: We thank the reviewer for the positive comments on our manuscript, and their helpful suggestions to include additional data to improve the paper.

Reviewer 3:

The reduced prevalence of insulin resistance and type 2 diabetes (T2D) in countries with endemic parasitic worm infections suggests a protective role for worms against metabolic disorders, however clinical evidence has been non-existent. This study investigated this hypothesis. Overall comments: This is a fun and novel approach. I like it.

However, I do have some concerns which should be further discussed.

Since exercise and energy intake also have pronounced effect of insulin resistance, could this affect the results? See eg: Healthy weight loss maintenance with exercise, liraglutide, or both combined JR Lundgren, C Janus, SBK Jensen, CR Juhl, LM Olsen, RM Christensen, ...New England Journal of Medicine 384 (18), 1719-1730 and Exploratory analysis of eating-and physical activity-related outcomes from a randomized controlled trial for weight loss maintenance with exercise and liraglutide single or...SBK Jensen, C Janus, JR Lundgren, CR Juhl, RM Sandsdal, LM Olsen, ...Nature Communications 13 (1), 4770

Response: We completely agree with the reviewer that exercise and energy intake were potential confounding variables for monitoring insulin resistance in this study. To address this, we asked participants to regularly complete exercise and food questionnaires, with the primary goal of addressing whether any of these habits changed after randomisation and treatment, which could explain improvements in metabolic health. Data already included in the paper (Supplementary Figures 2A-2B) quantify dietary habits (PREDIMED score) and Physical Activity (Metabolic Equivalents of Task/week). These data show that dietary habits did not substantially or consistently change in people from the Placebo or Hookworm-treated groups, suggesting that is unlikely that improvements in insulin resistance seen following hookworm infection were due to improved diet alone. Similarly, weekly exercise habits were not consistently changed in any treatment group (if anything, people were generally exercising less), again suggesting that improvement in insulin resistance were not likely due to increased exercise alone. We have added additional text in the revised Discussion (Page 18, line 334-336) to make this clearer, and cited one of the publications mentioned by the reviewer (Page 9, line 182-183) to illustrate why it was important to carefully monitor exercise and diet to minimise these confounding variables.

Did the authors consider the health care systems and the level of testing for T2D in countries with worm infections? Please see eg. Health care professionals from developing countries report educational benefits after an online diabetes course NJ Wewer Albrechtsen, KW Poulsen, LØ Svensson, L Jensen, JJ Holst, BMC medical education 17, 1-8

Response: We thank the reviewer for this question. We feel that this is of most relevance when trying to understand the inter-twined public health-related issues caused by Type 2 diabetes in helminth-endemic countries. For example, it remains possible that discrepancies in the level of testing for worm infections and Type 2 diabetes between different countries may have complicated the epidemiological or meta-analyses studies we have cited, which reported that people with worm infections were less likely to have metabolic dysfunction.

However, given our focus is to report clinical trial results of a potential new treatment for Type 2 diabetes, we feel that matters relating to epidemiology, education and health care systems are not integral for the overall message of the manuscript.

Did the authors consider the minimum relevant effect size between groups?

Response: In reporting these study outcomes, we have only reported the actual effect sizes between treatment groups based on our data. We feel that our small proof of principle study is not adequately powered to inform a minimum relevant effect size, which may be more relevant when planning a larger clinical study that is powered to determine efficacy.

REVIEWERS' COMMENTS

Reviewer #1 (Remarks to the Author):

The authors have addressed my previous comments and I am happy with the changes. I only have a few minor points that could improve the clarity of the manuscript:

1. PAGE 6 line 101 add "three-arm": "We conducted a 2-year Phase Ib randomised, double-blinded, three- arm placebo-controlled trial"

2. PAGE 19 line 365 "... higher worm doses and faecal EPGs were not associated with .." non-significant result does not imply there is no association. It would be better to say that correlation analyses didn't show a significant association.

Similarly PAGE 20 line 385 "Interestingly, hookworm infection did not improve blood lipid" better to say "didn't show a significant improvement .."

Reviewer #2 (Remarks to the Author):

The authors have done a great job addressing the concerns from the first submission. This is an exciting study that will have a high impact on the field.

Reviewer #3 (Remarks to the Author):

My concerns were sufficiently addressed. I no longer have concerns.

A point-by-point response to Reviewer 1, immediately below:

- 1. PAGE 6 line 101 add "three-arm": "We conducted a 2-year Phase Ib randomised, double-blinded, three- arm placebo-controlled trial"**

Response: This has been added on page 6 line 104

- 2. PAGE 19 line 365 "... higher worm doses and faecal EPGs were not associated with .." non-significant result does not imply there is no association. It would be better to say that correlation analyses didn't show a significant association.**

Response: We agree with the reviewer and have amended text on page 15, line 325 accordingly.

- 3. Similarly PAGE 20 line 385 "Interestingly, hookworm infection did not improve blood lipid" better to say "didn't show a significant improvement .."**

Response: We have amended text on page 16, line 345 accordingly.